# Crosstalk between MicroRNA and Oxidative Stress in Primary Open-Angle Glaucoma

**DOI:** 10.3390/ijms22052421

**Published:** 2021-02-28

**Authors:** Saray Tabak, Sofia Schreiber-Avissar, Elie Beit-Yannai

**Affiliations:** Department of Clinical Biochemistry and Pharmacology, Ben-Gurion University of the Negev, Beer-Sheva 84105, Israel; sarayt@post.bgu.ac.il (S.T.); sofia@bgu.ac.il (S.S.-A.)

**Keywords:** primary open angle glaucoma, oxidative stress, trabecular meshwork, intraocular pressure, miRNA, aqueous humor, retinal ganglion cells

## Abstract

Reactive oxygen species (ROS) plays a key role in the pathogenesis of primary open-angle glaucoma (POAG), a chronic neurodegenerative disease that damages the trabecular meshwork (TM) cells, inducing apoptosis of the retinal ganglion cells (RGC), deteriorating the optic nerve head, and leading to blindness. Aqueous humor (AH) outflow resistance and intraocular pressure (IOP) elevation contribute to disease progression. Nevertheless, despite the existence of pharmacological and surgical treatments, there is room for the development of additional treatment approaches. The following review is aimed at investigating the role of different microRNAs (miRNAs) in the expression of genes and proteins involved in the regulation of inflammatory and degenerative processes, focusing on the delicate balance of synthesis and deposition of extracellular matrix (ECM) regulated by chronic oxidative stress in POAG related tissues. The neutralizing activity of a couple of miRNAs was described, suggesting effective downregulation of pro-inflammatory and pro-fibrotic signaling pathways, including nuclear factor kappa-light-chain-enhancer of activated B cells (NF-kB), transforming growth factor-beta 2 (TGF-β2), Wnt/β-Catenin, and PI3K/AKT. In addition, with regards to the elevated IOP in many POAG patients due to increased outflow resistance, Collagen type I degradation was stimulated by some miRNAs and prevented ECM deposition in TM cells. Mitochondrial dysfunction as a consequence of oxidative stress was suppressed following exposure to different miRNAs. In contrast, increased oxidative damage by inhibiting the mTOR signaling pathway was described as part of the action of selected miRNAs. Summarizing, specific miRNAs may be promising therapeutic targets for lowering or preventing oxidative stress injury in POAG patients.

## 1. Introduction

Oxidative stress is generated by the imbalance between the production and accumulation of reactive oxygen species (ROS) and reactive nitrogen species (RNS) in cells and tissues [1], and the efficacy of antioxidant defenses to detoxify them [2]. ROS and RNS are generated as by-products of oxygen and nitric oxide metabolism, and consist of superoxide anion, hydrogen peroxide, hydroxyl radicals, and peroxynitrite. Both ROS and RNS have a dual role as being either useful by participating in cell signaling pathways, or harmful to the living system by promoting pathological processes such as inflammation, fibrosis, and apoptosis. Environmental stressors (UV, ionizing radiations, pollutants, smoke, heavy metals, and xenobiotics) or endogenous stressors (the mitochondrial electron transport chain and oxidative burst of phagocytes) greatly contribute to the increase in ROS and RNS production. As RNS are produced under ROS exposure, we will use the term ROS through the review for simplicity. Under physiological homeostasis, ROS participate in a series of cell signaling, which are essential for the cell’s existence. Increased intracellular oxidant levels in specific cells or tissues, above homeostasis levels, leads to two main effects: direct damage to diverse cell components including proteins, nucleotides, and lipids, and secondly, activation of specific signaling pathways leading to morphological damage and cellular functional weakness. These effects may influence numerous cellular processes related to the development of age-related diseases [3,4]. Cells display an antioxidant defensive system based on two arms, the dominant is comprised of enzymatic elements, for example: superoxide dismutase (SOD), catalase, and glutathione peroxidase [5]. The second arm involves waste products such as uric acid; active proteins, for example, albumin, cell origin molecules such as glutathione, and vitamins absorbed from the diet for example ascorbic acid, tocopherol, and others all known as low molecular weight antioxidants (LMWA).

Oxidative stress is responsible for developing and accelerating ocular diseases, including glaucoma disease [6]. Glaucoma is a chronic, degenerative optic neuropathy, damaging the optic nerve head [7] that is characterized by progressive degeneration of retinal ganglion cells (RGC) with a specific pattern of changes in the optic nerve head and retinal nerve fiber layer [8]. Glaucoma is divided into primary open and closure angle glaucoma and secondary glaucoma which can result from trauma, medication, and tumor or pseudo-exfoliation glaucoma [7]. Primary open-angle glaucoma (POAG) is a progressive optic neuropathy and the prominent cause of irreversible blindness [9]. Increased intraocular pressure (IOP) is a major risk factor for POAG [10], but additional factors that may affect the eye were shown to play a significant role, namely, increased glutamate levels [11], alterations in nitric oxide (NO) metabolism [12], vascular alterations [13,14], and ROS-associated oxidative damage [15,16]. Mutations of specific genes [17,18,19,20] and mechanical stress [21] due to elevated IOP are also important factors for disease progression. The damage related to IOP is expressed by the occurrence of degenerative phenomena that affect the sclero-corneal trabecular meshwork (TM) [22], the epithelium responsible for aqueous humor (AH) drainage from the eye’s anterior chamber [23]. Human TM formed by collagen lamellae lined by endothelial cells is abundant in the extracellular matrix (ECM), filling the gaps between the lamellae through which the AH passes. The maximum resistance to the AH outflow is situated at the periphery of the juxtacanalicular tissue, connected functionally and anatomically to the Schlemm’s Canal. The resistance of AH drainage through this pathway results in IOP increase and TM degeneration [24]. Studies suggest that oxidative DNA damage accumulates in this degenerating TM, accelerating a neuroinflammation process which drives the neurodegeneration in POAG pathology [25,26].

During recent years, the increasing investigation of control mechanisms of the gene–environment interactions led researchers to assume that microRNAs (miRNAs) are molecular mediators that participate in the regulation of oxidative stress and ROS pathways. miRNAs are formed by short single-stranded nucleotides (18–23 bp length) [27] that bind specific sequences within their target messenger RNA to regulate the expression of specific genes at the post-transcriptional level [28]. Growing evidence supports the role of miRNAs in critical/physiological cellular processes, such as oxidative stress [29], regulated by the pathophysiology of different disorders, including POAG [27]. Therefore, miRNAs detected in AH [30,31] represent new candidate biomarkers for the diagnosis, classification, prognosis, and responsiveness to treatment [27,29]. Nevertheless, the detailed mechanisms of action of miRNAs are not yet fully elucidated. In this review, we will focus on the contribution of oxidative stress to POAG pathological conditions, mediated by different types of machinery, factors, target genes, and specific miRNAs for seeing potential treatment strategies targeted at multiple signaling pathways or pathological components.

## 2. The Role of TM Cells in IOP Regulation and Oxidative Stress Occurrence

The TM is the key component of the AH outflow pathway, contributes to the majority of outflow resistance, and therefore, regulates IOP. In POAG, the TM undergoes a series of pathologic changes, causing increased outflow resistance and elevated IOP [32]. Elevated IOP compresses the structure in and around the optic nerve head, disturbing the axoplasmic transport within the nerve fibers. This leads to the death of retinal ganglion cells and their axons, resulting in thinning of the neuro-retinal rim and excavation of optic nerve head [33]. IOP increase is related to oxidative degenerative processes affecting the TM and specifically its endothelial cells. Then, ROS reduce local antioxidant activities, inducing outflow resistance and exacerbating the activities of superoxide dismutase and glutathione peroxidase in glaucomatous eyes. In this context, in vivo study revealed that lower systemic antioxidant capacity measured by ferric-reducing activity was involved in the pathogenesis of POAG via IOP elevation [34]. Furthermore, hydrogen peroxide induces rearrangement of TM cells and compromises their integrity [35].

## 3. Oxidative Stress Effects on POAG Related Tissues and Mechanisms

Exposure to sunlight and high oxygen concentration lead to a higher oxidative stress burden in the eye than other tissues, which can be further complicated by additional oxidative stressors [36]. With regards to the POAG pathology, we will focus on the main two tissues influenced by oxidative stress damage, the TM [9] and RGC [37]. The accumulation of ROS and the immune-stimulatory signaling enhanced by oxidative stress seem to result from the combination of TM tissue malfunction in the conventional outflow pathway and the neuroinflammation process [38]. Elevation of oxidative stress-related markers, low antioxidant resistance, dysfunction/activation of glial cells, activation of the nuclear factor kappa-light-chain-enhancer of activated B cells (NF-κB) pathway, and the up-regulation of pro-inflammatory cytokines are all related to the development of POAG [25]. Promoting the matrix metalloproteinases’ (MMPs) expressions responsible for ECM degradation by activated NF-κB in the initial stage of glaucoma [39] is meaningful in lowering IOP. Presently, IOP is the only risk factor affected by medication or glaucoma surgery [8]. Hence, a new approach to POAG treatment should be considered.

## 4. Oxidative Stress and Mitochondrial Dysfunction in POAG Pathology

With aging, there is a reduction in the antioxidant network functions, which results in oxidative damage accumulation in cells and tissues, and a higher susceptibility to morbidity and mortality [40,41]. Mitochondria contribute to aging through the accumulation of mitochondrial DNA (mtDNA) mutations, and the production of ROS. Mitochondrial matrix enzymes, the α-keto acid dehydrogenase complexes [42], the mitochondrial electron transport chain, and the loss of mitochondrial ability in buffering Ca^2+^ [43] are all factors that stimulate ROS production in the mitochondria, resulting in cell death via apoptosis or necrosis [44]. In POAG, the accumulation of excessive ROS can induce TM damage, which results in conventional outflow pathway defects [25] and exacerbates the injury to the optic nerve head and RGC [38]. As high metabolism occurs in RGC, proper mitochondrial function is essential for these neurons that die in glaucoma [45,46]. Besides, mtDNA changes and a decrease in the mitochondrial respiratory activity, related to mitochondrial abnormalities, are more common features than genetic mutations of related POAG genes, such as MYOC and OPTN [47]. Moreover, in vivo experiments in humans revealed that both IOP increase and visual field reduction are significantly related to the amount of oxidative DNA damage affecting TM cells [48].

## 5. Oxidative Stress-Related TM Damage

The TM is the most sensitive tissue of the anterior segment of the eye and is prone to oxidative stress [49]. In glaucoma patients’ TM, significant levels of 8-oxo-20-deoxyguanosine (8-OH-dG) [26], HSP72 [41], and glutamine synthetase [50] were found, indicating DNA oxidative stress damage, stress, and excitotoxicity-related protein expression, respectively.

When TM cells are chronically exposed to oxidative stress, significant functional damage to their lysosome system has been reported. Liton et al. described the accumulation of nondegradable material resulting from diminished autophagy [51], accelerating cell senescence as measured by increased senescence-associated-β-galactosidase and senescence-associated secretory phenotype (SASP) protein expression/cellular levels [52,53]. These harmful processes contribute to the functional damage of TM tissue [52], which were shown by Guorong et al. to result from phenotypic changes altering TM tissue microenvironment and promoting age-associated pathological alterations [53].

For instance, molecular changes in POAG were detected in glaucomatous human TM cells morphological analysis. These changes included ECM accumulation, cytoskeleton disruption, cell death, progressive senescence, NF-κB activation, and the release of inflammatory markers [39,54]. Interestingly, collagen type I accumulates ROS-scavenging residues (Tyr/Phe/Met) to prevent mechano-oxidative damage to the tissue, meaning that mechanical stress on collagen leads to ROS production [55]. Under chronic stress conditions, the endoplasmic reticulum (ER) accumulates reactive oxygen species and promotes oxidative stress-induced TM damage due to its inability to act in response to unfolded or misfolded proteins [56,57]. The damaged ER activates inflammatory processes via NF-κB, mitochondrial changes, and enhanced TM cell apoptosis, which lead to elevated IOP [58], and activated glial cells, and N-methyl-D-aspartate (NMDA) and AMPA receptors [59].

Wang et al. proposed that the increased expression of the endothelial-leukocyte adhesion molecule (ELAM-1) in TM cells sustains the IL-1-induced pathogenic role of oxidative stress in POAG [60]. Elevation in ELAM-1 regulates the NF-kB factor to lower oxidative stress [48,60].

The highly adaptive complex and efficient antioxidant defense system of TM cells and AH include two antioxidant classes: enzymatic, such as glutathione peroxidase, glutathione [61,62], SOD [63], catalase [64], and nonenzymatic, such as ascorbic acid [65]. Nuclear factor erythroid 2-related factor 2 (Nrf2) is a transcription factor activated by oxidative stress; it binds to antioxidant response elements (ARE) that lead to a cellular antioxidant response [66]. Low levels of ROS induce antioxidant gene activation, related to the Nrf2 pathway, while medium levels activate NF-κB signaling and high levels lead to apoptosis or necrosis [67].

High levels of ROS and particularly hydrogen peroxide levels in TM cells reduce local antioxidant activities, which, in turn, increase AH outflow resistance thus exacerbating superoxide dismutase and glutathione peroxidase activities [35]. High levels of hydrogen peroxide affect the secretion of adhesion proteins to the ECM of TM cells, which result in cytoskeleton reorganization and cause inadequate adhesion of TM, eventually leading to cell loss [9].

## 6. AH Composition Alternations as Response to Oxidative Stress in POAG Patients

AH composition depends on the metabolites produced during its generation and those acquired during its passage through various anterior segment regions [68]. Since the proteomic AH profile of POAG patients is completely altered compared to healthy individuals [59], it is significant to investigate a large number of pro-/anti-oxidation agents that are found in the AH of POAG patients exposed to oxidative stress. An increase in NO levels, endothelin 1 (ET-1) [69], hydroxyproline (derived from collagen hydrolysis) [70], and acetate (regulates outflow dynamics, due to either cell loss or the dysfunction of sub-cellular structures) increase POAG AH. Both levels of plasminogen activator inhibitor-1 (PAI-1) and transforming growth factor-beta 2 (TGF-β2) are also elevated in POAG patients [69,70,71]. Furthermore, the increase in transthyretin (TTR), prostaglandin H2 D-isomerase (PGDS), and caspase 14 in POAG AH can lead to TM apoptosis [71]. While a significant decrease in the antioxidant activity in the AH of POAG patients was detected, alternations of SOD, glutathione peroxidase, catalase, and MDA activities were noticeable [15,37,63]. The presence of specific proteins in AH, such as junction proteins, chains, and cadherins, which under physiological conditions contribute to tissue integrity, and determine both RGC and TM damage degree, was also affected [72].

## 7. Oxidative Stress and RGC Damage in POAG Patients

Feilchenfeld et al. suggested that low perfusion pressure that compromises ocular blood flow auto-regulation is affected by vascular insufficiency [73]. Such a condition of sustained hypoxic insult promotes an increase in glial activity, immune system involvement, and IOP [38]. ROS overproduction, lack of ATP supply, mitochondrial function interruption, high Ca^2+^ traffic across the neuronal membranes, increased lipid peroxidation and protein carbonyl content are recognized features of neuronal apoptosis that lead to different neurodegenerations in RGC [41,43,73,74,75,76]. Exogenous application of ROS was found to trigger in vitro RGC apoptosis via a caspase-independent receptor and mitochondrial pathways [77,78]. The accumulation of advanced-glycation-end-products (AGEs) that lead to ROS generation was found in the glaucomatous retina and optic nerve head [79]. The advanced glycation process may be related to the activation of signaling molecules (mitogen-activated protein kinases, MAPKs, or NF-kB) linked to oxidative stress in glaucoma. Neurodegenerative injury and glial activation in the course of glaucomatous degeneration enable tissue healing by evoking an immune response that restores tissue homeostasis. Nevertheless, oxidative stress and aging-related components may induce a malfunction in the regulation of innate and adaptive immune response and act as a path for transforming the beneficial immunity into a neuroinflammatory degenerative process that results in elevated production of pro-inflammatory molecules TNF-α, NF-κB, nitric oxide synthase, and cyclooxygenase-2 [80,81]. This, in turn, will contribute to the formation of ROS and RNS while creating a cycle of responses that aggravate the condition of the cells and relevant tissues.

## 8. Functional Roles of Specific miRNAs Found in the Aqueous Humor Related to POAG

Genetic susceptibility is a crucial factor in POAG, reflected in miRNA expression and function alternations, thereby leading to POAG occurrence and development. A large number of miRNAs are involved in the regulation of IOP and play a crucial role in the increase in IOP found in the majority of POAG patients. Additionally, factors such as mechanical stress, hypoxia, and inflammation were shown to interfere with optic nerve head damage through miRNA [27]. Various miRNAs are abundantly expressed in the human eye and have a clear functional disposition to be used as biomarkers to assist in the early diagnosis of POAG [82,83,84]. Several miRNAs were mentioned as POAG treatment points of interest [85].

miR-29b inhibits collagen I, III, IV synthesis, causing ECM deposition in the TM, and has an anti-fibrotic effect as was shown in human, rat, and in vitro models [83,84,86,87,88]. miR-29b is an activator of the Wnt/β-Catenin [89,90] and PI3K/Akt/Sp1 [88,91] signaling pathways. The Wnt/β-catenin signaling pathway regulates cell–cell adhesion and ECM expression, while the expression of collagen type I is inhibited by this miRNA through the PI3K/Akt/Sp1 signaling pathway.

miR-182 is abundantly expressed in the mammalian retina and is necessary for optic nerve development [92,93,94]. An increase in miR-182 expression levels was reported in the AH and TM cells of glaucoma patients [93,94,95]. miR-182 also has antioxidant and anti-inflammatory effects and is responsible for the enhancement of SOD activity [95]. miR-182 protects RGC from oxidative stress damage [96] and inhibits the activation of microglia by targeting Toll-like receptor 4 (TLR4), which activates retinal inflammation as shown in rat and human cell models [97].

miR-141 prevents the apoptosis of TM cells and RGC. It down-regulates the expression level of PTEN (phosphatase and tensin homolog) [98] through the PI3K/Akt/mTOR pathway [99], and thereby promotes cell proliferation and inhibits cell apoptosis. Moreover, miR-141-3p participates in oxidative stress regulation and inhibits N-methyl-D-aspartate (NMDA)-induced mouse RGC by inhibiting MAPK signaling [100].

miR-27a exerts a protective role on human TM cells under hydrogen peroxide administration. Actually, Salidroside (a strong antioxidant) activates the PI3K/AKT and Wnt/β-catenin pathways through the enhancement of miR-27a expression in H_2_O_2_-injured human TM cells [101].

Another miRNA also involved in glaucoma is miR-17-5p. In human TM cells, this miRNA has a role in regulating proliferation and apoptosis in response to oxidative stress [102].

In POAG patients, neuroinflammation is an important mechanism underlying optic nerve injury [103]. miR-155 and miR-146a are expressed in activated immune cells and are essential for B cells immune response, macrophages, and microglia activation [104,105,106], which promotes inflammation [107]. In contrast, the main function of miR-146a is to inhibit inflammation and T cell adhesion [108,109,110]. Both miRNAs regulate the pro-inflammatory NF-κB signaling pathway. While miR-155 promotes NF-κB activation [108], miR-146a inhibits IL-1 receptor-associated kinase 1 and tumor necrosis factor receptor-associated factor 6 and inhibits inflammation [109].

Neuronal differentiation and anti-inflammatory effects of miR-124 treatment were shown to modulate the polarization of activated microglia and protect neurons in various ways [110]. Therefore, reducing the susceptibility of RGC to apoptotic stimuli may have the potential to strengthen medical effects by neuroprotective agents [111]. Long noncoding RNA (lncRNA) is a typical non-coding RNA, participating in regulating the transcription and translation of genes [112]. Emerging evidence has suggested the critical role of lncRNAs in the occurrence of POAG. These lncRNAs able to favor either the repression or expression of target mRNAs by sharing miRNA response elements with mRNA and then can alleviate the inhibition of the miRNA-mediated target gene. Due to the annotated functions of miRNAs involved in the pathogenesis of POAG, it is essential to discuss the contribution of cross-regulation between miRNAs and lncRNAs to cellular, physiologic, and pathologic processes. Yoon J. et al. summarized in their work various examples of direct cross-regulation among lncRNAs and miRNAs [113]. Concerning POAG, it was documented that CDR1as (antisense to the cerebellar degeneration-related protein 1 transcript) is a lncRNA responsible for the repression of miR-7, while Sry (sex-determining region Y) lncRNA serves as a sponge for miR-138.

## 9. Mechanisms of miRNAs in POAG

As was previously mentioned, the main reason for IOP elevation in POAG is the increased resistance to AH drainage through the TM, characterized by abnormal ECM deposition. Oxidative stress and TGF-β are the main triggers of the expression of ECM components. Changes in miRNAs are designed to enhance the viability and resist hypoxic attack accompanied by the oxidative stress response. These beneficial effects are related to various miRNAs, including miR-29b.

TGF-β2 promotes fibronectin expression [114] through miR-29b inhibition (increases collagen types I and IV) and the SMAD signaling pathway [115]. ROS in TM cells induces ECM deposition, which promotes cellular senescence, injury, and apoptosis [116,117]. miR-29b is significantly down-regulated in TM cells, associated with ECM deposition caused by oxidative stress. Furthermore, miR-29b and miR-24 were found to be involved in gene regulation in TM cells [118,119]. Additional anti-oxidation miRNAs, such as miR-182, miR-187, and miR-126, are involved in optic nerve injury caused by ischemia and hypoxia and are down-regulated in RGC [96]. Hypoxia down-regulates miR-126, increasing the expression of MMP-9, which aggravates RGC injury. Downregulation of miR-100 also has a protective effect against oxidative stress in RGC, protecting them from apoptosis by activation of the AKT/ERK and tyrosine kinase receptor (TrkB) pathways through phosphorylation [119,120]. Researchers have identified several miRNA processes taking place in neuronal homeostasis under different pathologies. miR-338, miR-7-5p, and miR-138 are involved in optic nerve damage caused by mechanical stress. Specifically, miR-338 regulates axon respiratory function and neurotransmitter uptake by inhibition of cytochrome C oxidase IV and ATP synthase (ATP5G1) mRNAs, encoded by mitochondrial genes [121,122]. miR-7-5p is involved in the electrical signal transduction of neurons through axons [123]. miR-138 inhibits nerve fiber demyelination during crush injury [124,125]. Recent findings demonstrate that miR-200c regulates TM cell contraction, and, as such, it contributes to IOP lowering in vivo and in vitro [119,126]. The mechanism of miR-200c action is based on post-transcriptional inhibition of genes associated with contraction regulation of TM cells, including Zinc finger E-box binding homeobox 1 (ZEB1) and 2 (ZEB2), forming homology 2 domain containing 1 (FHOD1), lysophosphatidic acid receptor 1 (LPAR1/EDG2), endothelin A receptor (ETAR), and Rho-A kinase. Pro-inflammatory cytokines such as TNF-α and interleukins are significantly up-regulated in the AH and retina in POAG [81,127,128,129] and are responsible for microglial activation and lymphocyte infiltration [130,131,132]. These findings emphasize the relation between inflammation and optic nerve injury in POAG. miR-182, miR-27a, miR-155, miR-146a, and miR-125b are all involved in optic nerve injury caused by inflammation. Activation of retinal local microglia that promote RGC apoptosis [130] and activation of the TLR4 pathway [133,134] both contribute to high IOP; hypoxia and ROS are the main causes of inflammation [103]. Up-regulation of miR-27a found in the retina of rats with elevated IOP [135], together with miR-182, inhibits inflammation by targeting TLR4 [97,136]. miR-155, miR-125b, and miR-146a regulate the activation of microglia in the inflammatory process [137,138]. Only miR-155-5p and miR-125b-5p are down-regulated in the AH of POAG patients [30,139].

IOP, retinal vascular perfusion pressure, and cerebrospinal fluid pressure are present around the retina, causing damage to the optic nerve [140]. In POAG, mechanical damage caused by elevated IOP deteriorates the optic nerve due to axonal transport failure in RGC [141]. Although not fully clear, it is currently accepted that in the early stage of hypoxia, there is a transient release of ROS contributing to oxidative stress. It was suggested that hypoxia increases ROS levels by activating the NADPH and xanthine oxidase pathways [142]. Oxidative stress and miRNAs that promote ROS are up-regulated by hypoxia [143]. For instance, in the early stage of hypoxia, the Hypoxia-inducible factor 1α (HIF-1α) inhibits apoptosis by up-regulation of miR-21 [144] and miR-210 [145]. Other results showed that miR-155 [146] and an increase in miR-210 levels [147] weaken the adaptive response and promote RGC apoptosis.

## 10. The Role of Oxidative Stress and Related miRNAs in POAG Physiology and Pathology

It is a well-known fact that increasing specific gene expression may potentially affect the physiology of AH outflow pathway by contributing to a larger deposition of collagen and other ECM components in the TM. Nevertheless, the potential involvement of miRNAs in the alterations in ECM synthesis induced by oxidative stress in TM cells has not been broadly investigated.

One of the prominent findings is related to miR-29b [116]. It was shown that miR-29b increased TM cell viability under chronic oxidative stress and physiologic oxygen concentrations. At physiological conditions, miR-29b negatively regulated the expression of collagens (COL1A1, COL1A2, COL4A1, COL5A1, COL5A2, COL3A1), laminin (LAMC1), and fibrillin (FBN) involved in the synthesis and deposition of ECM in TM cells. Under chronic oxidative stress conditions, miR-29b down-regulation resulted in increased expression of these genes. One of the logical and proven explanations for the results is that miR-29b negatively modulated the expression of collagens and other key components of the ECM in TM cells and decreased cytotoxicity in the presence of chronic oxidative stress through NF-kB regulation. It was suggested in the same study that NKRAS2, a negative modulator of the pro-inflammatory and pro-apoptotic factor NF-kB, was down-regulated by miR-29b. Additional papers reinforcing these findings are Wenying Ran et al. [148] and Mingxuan Wang et al. [149]. The decisive conclusion of their work was that up-regulation of Nrf2 protects TM cells and the RGC cells from the effects of TGF-β2 and fibrosis, caused by oxidative stress damage, by up-regulating miR-29b. An additional miRNA with a beneficial physiological effect is miR-141. miR-141 reduces UV light-induced oxidative stress via the activation of the Keap1- Nrf2 signaling pathway [150]. miR-93, on the other hand, is elevated in POAG pathology as a response to oxidative stress, inhibits cell viability, and induces apoptosis of glaucomatous TM via Nrf2 suppression [151].

Examination of Salidroside, a phenolic natural product with pharmacological effects in human TM cells exposed to oxidative stress, revealed that this type of glucoside of tyrosol can protect human TM cells against hydrogen peroxide evoked oxidative damage by activation of the PI3K/AKT and Wnt/β-Catenin pathways through enhancement of miR-27a expression [101].

In another study [152], the pleiotropic effects of *Lycium barbarum* polysaccharides (LBPs) on injured human TM cells as a result of exposure to hydrogen peroxide were investigated. LBPs significantly promoted cell viability by reducing apoptosis, cleaved-caspase 3/9, and ROS levels in TM cells after hydrogen peroxide administration. Hydrogen peroxide stimulation down-regulated the protein levels of p-PI3K and p-AKT, while LBPs countered the down-regulation and resumed the activation of the PI3K/AKT signaling pathway. The protective effect of LBPs, expressed via PI3K/AKT signaling activation, was reversed by miR-4295 inhibition. These results indicate that up-regulation of miR-4295 in human TM cells has a protective effect against oxidative damage.

Oxidative injury of human TM cells was enhanced by miR-7 through mTOR and MEK/ERK pathways’ down-regulation [153].

So far, the involvement of miRNAs in exposed TM cells to oxidative stress was presented. We will now focus on RGC cells and their corresponding expression of miRNAs.

miR-182 is a good example of RGC regulation in glaucomatous patients exposed to oxidative stress. The increase in the antioxidant SOD and decrease of cytochrome C release from mitochondria were regulated through miR-182 in hydrogen peroxide-treated RGC [96]. These results shed light on the role of miR-182 as an anti-oxidative and anti-apoptotic agent suppressing the mitochondrial apoptotic pathway.

Overexpression of miR-26a protects RGC cells against cytotoxicity and apoptosis induced by hydrogen peroxide through down-regulation of PTEN and phosphorylation of AKT protein downregulation [154].

The last one to be discussed is miR-124. miR-124 prevents oxidative stress and apoptosis in human lens epithelial cells by suppressing the activation of the NF-κB pathway [155]. However, other tissues influenced by oxidative stress damage in glaucomatous individuals were not examined (Table 1).

## 11. Concluding Remarks and Future Perspectives

POAG targets a variety of different tissues located in both anterior (TM cells) and posterior (RGC and optic nerve head) ocular segments. These tissues are highly exposed to oxidative stress, expressed in neurodegenerative and inflammatory disorders, leading to cell injury, apoptosis, AH outflow resistance, elevation of IOP, and finally to visual field loss.

Slowing disease progression and preservation of quality of life are the main goals for glaucoma treatment, but these do not always succeed in stopping the gradual worsening of visual function, and some patients continue to lose vision despite all currently available treatments [8].

Another approach to treat POAG patients suggests using hyperbaric oxygen therapy that exposes the eye to increased oxygen concentration. miRNAs are stable, not degraded with ease, however, the higher oxygen concentration in the AH and the risk of damage to TM cells may be greater [159].

miRNAs are stable, not degraded with ease [160], can be stored for a long time, and most importantly, specifically regulate the expression of target proteins at the post-transcriptional level. The variety of miRNAs that exist in the AH is beneficial for the early diagnosis of POAG [139,161,162]. Thus, it should be further investigated whether regulation of ROS levels, mediated by miRNAs, can protect against POAG progression. This review demonstrated the favorable role of miRNAs as diagnostic and therapeutic tools for POAG. We report on five prominent miRNAs that participate in the regulation of oxidative stress effects in POAG pathology; miR-29b, miR-27a, and miR-124 all protect TM cells against oxidative damage induced by the expression of NF-kB, by inhibition of the main signaling pathways: TGF-β2, PI3K/AKT, and Wnt/β-Catenin. miR-29b and miR-27a prevent ECM deposition, while miR-124 mainly prevents apoptosis. miR-182 increases SOD levels in RGC and suppresses mitochondria dysfunction through negative NF-kB downregulation. Unlike miR-29b, miR-27a, miR-124, and miR-182, which stand out due to their characteristics as antioxidants and anti-apoptotic agents, miR-7 is a pro-modulator of oxidative injury through mTOR and downregulation of MEK/ERK pathways (Figure 1). These data lead to some important conclusions. First, the expression of collagen type I in the ECM of glaucomatous patients exposed to oxidative stress may be regulated by specific miRNAs to limit ECM deposition to maintain normal levels of AH outflow facility. The prominent risk factor for ROS levels elevation is NF-kB; hence, a protective mechanism of mRNA targeting should include inhibition of TLR4 in TM and RGC tissues.

It is expected that this miR-29b function would enhance apoptosis under chronic oxidative stress conditions since p53 has pro-apoptotic effects [163,164]. As such, miR-29b can be an attractive target for interference in POAG. Since the action of TGF-β2 in POAG is largely mediated through miR-29b [165], further examination of alternation in miR-29b levels in POAG patients can be used as a tool for disease detection. Examination of relevant miRNAs in response to the antioxidant, Edaravone, which decreases apoptotic cell death, oxidative damage to DNA and lipids, and angiogenesis through inhibiting JNK and p38 MAPK pathways in glaucoma, can shed new light regarding facilitating therapy through MAPK pathway regulation [6]. Furthermore, the possible involvement of oxidative damage to DNA in POAG pathogenesis may indicate that DNA mutations are involved in a variety of different human diseases with miRNA treatment as a potential therapeutic strategy that should be investigated. Recent studies emphasized the importance of extracellular vesicles, and specifically exosomes, as protective signaling mediators in TM cells during oxidative stress [166]. IL-1β-induced acute neuroinflammation and oxidative stress resulted in the release of a specific subset of miRNAs via exosomes, potentially regulating the inflammatory response [167]. The capacity of extracellular vesicles to carry protective signals following oxidative stress is well documented [168,169,170]. Exosomes are bi-lipid layered membranous vesicles with a diameter of approximately 30–100 nm characterized by specific cell surface markers [171,172]. It has been reported that cell miRNAs reach the extracellular environment through exosomes and that the exosomal cargo of cellular proteins, lipids, and miRNAs play an important role as mediators of intercellular crosstalk between the producing and recipient cells [173]. A genome scan for miRNA-related genetic variants associated with POAG [118] and a comparison of miRNA expression in AH of normal and POAG individuals [31] were widely examined. Additionally, appropriate oxidative stress biomarkers in AH of POAG patients should be further examined, focusing on the AH-producing cells and the non-pigmented ciliary epithelium (NPCE) located in the anterior chamber of the eye, exiting the eye through the TM cells in the conventional pathway [30]. A potential therapeutic target for glaucoma can be achieved by suppression of miRNAs that are considered as pro-mediators of oxidative stress, such as miR-210, using long non-coding RNAs transcripts (lncRNAs) [117]. In conclusion, we suggest further investigating the role of POAG-related miRNAs as antioxidant machinery, examining their dual role as pro- or anti-inflammatory/apoptotic agents as a response to diverse concentrations of ROS in TM cells and RGC.

## Figures and Tables

**Figure 1 ijms-22-02421-f001:**
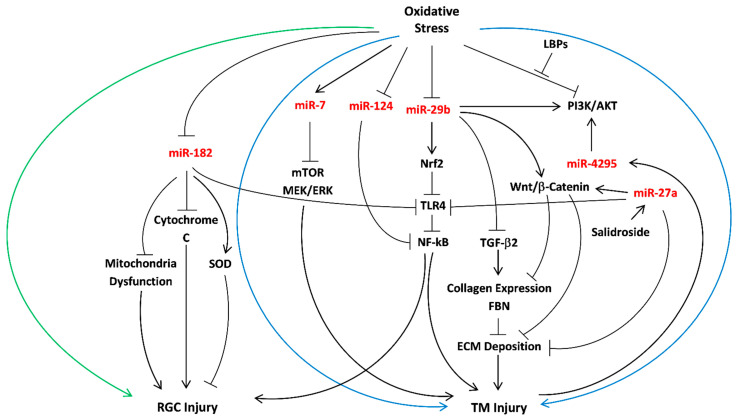
Overview of important miRNAs as mediators of oxidative stress in POAG. Summary of the miRNAs (red) involved in the protective mechanisms against oxidative stress in POAG. miR-7, miR-24, miR-27a, miR-29b, miR-4295 have an anti-oxidant effect in TM cells (surrounded by the blue arrows), while miR-182 reduces oxidative stress damage in RGC (surrounded by the green arrow). The following abbreviations refer to TM (trabecular meshwork), RGC (retinal ganglion cells), LBPs (*Lycium barbarum* polysaccharides), FBN (fibrillin), ECM (extracellular matrix), SOD (superoxide dismutase), TGF-β2 (transforming growth factor-beta 2), NF-kB (nuclear factor kappa-light-chain-enhancer of activated B cells), TLR4 (toll-like receptor 4), and Nrf2 (nuclear factor erythroid 2-related factor 2).

**Table 1 ijms-22-02421-t001:** Mechanisms of microRNAs (miRNAs) in primary open-angle glaucoma (POAG).

miRNA	Mechanisms	Site Effecting	Pathway Involved	Reference
**miR-7**	Mechanical Stress	Optic Nerve, TM Cells	Electrical Signal Transduction, mTOR, MEK/ERK	[125]
**miR-21**	Hypoxia	ECM stiffness RGC, Corneal Epithelium	TGFβ1, HIF-1α	[85,131,156]
**miR-24**	ECM deposition	TM Cells	TGF-β	[118]
**miR-26a**	Cytotoxicity, Apoptosis	RGC	PTEN, AKT	[154]
**miR-27a**	Inflammation, Hypoxia	Optic Nerve, Retina, RGC, TM Cells, AH	Activation of retina local microglia, TLR4, PI3K/AKT, Wnt/β-Catenin	[97,100,103,138,142,144,153]
**miR-29b**	Cell senescence, Injury, Apoptosis, ECM deposition, Fibrosis	TM Cells, AH	NF-kB, NKRAS2, Nrf2, TGFβ2, Collagens, LAMC1, FBN, Wnt/β-Catenin, PI3K/Akt/Sp1	[86,87,88,98,115,116,117,118,148,149]
**miR-93**	Apoptosis	TM Cells	NF-kB, Nrf2	[151]
**miR-100**	Apoptosis, Neuronal Growth	RGC	AKT/ERK, TrkB	[119,120]
**miR-124**	Inflammation, Apoptosis	AH	Activation of microglia, NF-κB	[110,156]
**miR-125b**	Inflammation, Hypoxia	Optic Nerve, RGC	Activation of microglia, TLR4	[103,138,141,142,145,146]
**miR-126**	Ischemia, Hypoxia	Optic Nerve, RGC	MMP-9	[96]
**miR-138**	Mechanical Stress	Optic Nerve	Nerve Fiber Demyelination	[127,128]
**miR-141**	Apoptosis	TM Cells, RGC, AH	PTEN, PI3K/Akt/mTOR, MAPK	[98,99,100]
**miR-146a**	Inflammation, Hypoxia, Immune response	Optic Nerve, RGC, AH	Activation of microglia and macrophages, TLR4, NF-κB	[103,107,109,138,141,142,145,146,157,158]
**miR-155**	Inflammation, Hypoxia, Apoptoses, Immune response	Optic Nerve, RGC, AH	Activation of retina local microglia and macrophages, TLR4, NF-κB	[103,107,108,133,138,141,142,145,146]
**miR-182**	Ischemia, Hypoxia, Inflammation	Optic Nerve, RGC, AH	Cytochrome C, TLR4	[89,90,91,92,93,94,95,96,97,144]
**miR-187**	Ischemia, Hypoxia, Inflammation, Apoptosis	Optic Nerve, RGC	P2X7 Receptor	[96]
**miR-200c**	Mechanical Contraction	TM Cells	ZEB1, ZEB2, FHOD1, LPAR1/EDG2, ETAR, RHOA Kinase	[119,126]
**miR-210**	Hypoxia, Apoptosis	RGC	HIF-1α	[132,134]
**miR-338**	Mechanical Stress	Optic Nerve	Cytochrome C Oxidase IV, ATP5G1	[123,124]
**miR-4295**	Apoptosis	TM Cells	PI3K/AKT, caspase 3/9	[152]

TM-trabecular meshwork, ECM-extracellular matrix, RGC-retinal ganglion cells, mTOR-mechanistic target of rapamycin, MEK/ERK- mitogen-activated protein kinase/extracellular signal-regulated kinases, TGF- Transforming growth factor, HIF-Hypoxia induced factor, PTEN- Phosphatase and tensin homolog, AH-aqueous humor, TLR- toll-like receptors, PI3K- phosphoinositide 3-kinases, NF-kB- nuclear factor-kappa B, Nrf2-uclear factor erythroid 2–related factor 2, TGFβ- transforming growth factor, TrkB- tropomyosin receptor kinase B, LAMC- laminin subu-nit gamma, FBN- fibrillin, ZEB- zinc finger e-box binding homeobox, FHOD1- formin homology 2 domain contain-ing, LPAR-lysophosphatidic acid receptor, RhoA- ras homolog family member A.

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
