# Peer review of "Crosstalk between MicroRNA and Oxidative Stress in Primary Open-Angle Glaucoma"

_ijms, 2021, doi:10.3390/ijms22052421_

Round 1

Reviewer 1 Report

The authors described the relevant findings about the role of MicroRNA and Oxidative Stress in Primary Open-Angle Glaucoma.

The review is generally well written, complete and robust, and provides a timely update of the literature on miRNA and oxidative stress in primary open-angle glaucoma. The authors have summarized the published literature comprehensibly with illustrations and figures.

It has been  noted that in some paragraphs there is  informations  just reported in the reviews and papers of Saccà SC group, which are also mentioned in the references.

However, I suggest the following revision:

In section “Functional Roles of Specific miRNAs found in the Aqueous Humor Related to POAG” it is also essential to discuss the contribution of cross-regulation between microRNAs (miRNAs) and lncRNAs, which is able to favor either the repression or expression of target mRNAs.

Zhou, M.; Lu, B.; Tan, W.; Fu, M. Identification of lncRNA–miRNA–mRNA regulatory network associated with primary open angle glaucoma. BMC Ophthalmology 2020, 20, doi:10.1186/s12886-020-01365-5.

Yoon, J.-H.; Abdelmohsen, K.; Gorospe, M. Functional interactions among microRNAs and long noncoding RNAs. Seminars in Cell & Developmental Biology 2014, 34, 9–14, doi:10.1016/j.semcdb.2014.05.015.  

Author Response

 We would like to thank the reviewer for this comment. Indeed, LncRNAs are important and their roles in POAG should ne addressed. The two mentioned reference were used as the basis for the sentences added (lines 230-240) addressing this point (lines 230-240)

"Long noncoding RNA (lncRNA) is a typical non-coding RNA, participating in regulating the transcription and translation of genes [112]. Emerging evidence has suggested the critical role of lncRNAs in the occurrence of POAG. These lncRNAs able to favor either the repression or expression of target mRNAs by sharing miRNA response elements with mRNA and then can alleviate the inhibition of the miRNA-mediated target gene. Due to the annotated functions of miRNAs involved in the pathogenesis of POAG, it is essential to discuss the contribution of cross-regulation between miRNAs and lncRNAs to cellular, physiologic and pathologic processes. Yoon J. et al summarized in their work various examples of direct cross-regulation among lncRNAs and miRNAs[113]. Concerning POAG, it was documented that CDR1as (antisense to the cerebellar degeneration-related protein 1 transcript) is a lncRNA that responsible for the repression of miR-7, while Sry (sex-determining region Y) lncRNA serves as a sponge for miR-138."

Reviewer 2 Report

The authors provide a very interesting and useful review on microRNA and oxidative stress in primary open angle glaucoma.
The bibliography is exhaustive.

1° In the abstract, it
is excessive to say that there is a lack of effective method for POAG treatment. 80% of patients are well controlled with eye drops, surgery or laser.
It is nevertheless certain that new therapeutic targets can be developed.

It is not clearly enough reported that these oxydative stress anomalies do not constitute the primum movens of POAG.
I think that must be most clearly specify.

3° In fact, it is some time difficult to follow the
progression of oxidative stress on the different ocular tissues. However, the authors do not explain where it comes from and what is the primitive element causing oxydative stress and ROS increase in POAG. It is not always clear that the primum movens is the increase of IOP related and due to TM damage.
The authors explain that increase of ROS can induce TM damage which is clearly demonstrated.
But TM damage have to be present prior oxydative stress to explain increase of IOP and occurence of oxydatve stress. 4° It seems that some miRNAs can be considered as POAG treatment point of interest.
Especially miR-29b that can cause ECM deposition in the TM.
Is there any data which suggests that this miRNA could be initially abnormal in the POAG?

Author Response

1° In the abstract, it is excessive to say that there is a lack of effective method for POAG treatment. 80% of patients are well controlled with eye drops, surgery or laser.

It is nevertheless certain that new therapeutic targets can be developed.

The previous statement regarding "lack of effective methods for POAG" was changed to "Nevertheless, despite the existence of pharmacological and surgical treatments there is room for the development of additional treatment approaches."

2° It is not clearly enough reported that these oxidative stress anomalies do not constitute the primum moves of POAG.

I think that must be most clearly specify.

We rephrased the paragraph: The role of TM Cells in IOP regulation and Oxidative Stress Occurrence, so the relationship between oxidative stress and the developing damage to the ocular tissue is clarified (lines 78-95). Details regarding the physiological mechanisms accepted today to play a role in TM damage under oxidative stress were reviewed in paragraph Oxidative Stress and Mitochondrial Dysfunction in POAG Pathology (lines 111-124)

3° In fact, it is some time difficult to follow the progression of oxidative stress on the different ocular tissues. However, the authors do not explain where it comes from and what is the primitive element causing oxidative stress and ROS increase in POAG. It is not always clear that the primum moves is the increase of IOP related and due to TM damage.

The authors explain that increase of ROS can induce TM damage which is clearly demonstrated.

But TM damage have to be present prior oxidative stress to explain increase of IOP and occurrence of oxidative stress.

A number of mechanisms have been referred to the pathology of POAG mainly to explain RGC degeneration. The vascular theory (including: chronic intermittent ischemia, defective axon transport, trophic factor withdrawal, and loss of electrical activity) was proposed about forty years ago alongside an increasing number of publications focusing on oxidative stress involvement in the initiation of the POAG pathology. In our humble opinion there is no contradiction between these theories but rather there is room for the biological processes described in parallel when quite a few works have indicated cross-effects of the factors involved in oxidative stress on ischemic processes and vice versa. The fact that many glaucoma patients are classified as normal tension glaucoma, suggests that pathological processes associated with RGC injury are not exclusively dependent on intraocular pressure. However common stress-response markers, alpha B-crystallin and TIGR were expressed similarly in patients TM, suggesting common pathology pathways unrelated to IOP.

The present review focuses on oxidative stress and we describe the primary processes leading to oxidative damage to ocular tissue as follow: IOP increase is related to oxidative degenerative processes affecting the TM and specifically its endothelial cells.  ROS reduce local antioxidant activities, inducing outflow resistance and exacerbating the activities of superoxide dismutase and glutathione peroxidase in glaucomatous eyes. (line 84-85) The exposure to sunlight and high oxygen concentration lead to a higher oxidative stress burden in the eye than other tissues (line 99). We hope that this description gives proper expression to the pathological procedures involved in the development of glaucoma in terms of oxidative stress and damage to TM

4° It seems that some miRNAs can be considered as POAG treatment point of interest.

Especially miR-29b that can cause ECM deposition in the TM.

The reviewer is right and mir-29b was suggested as having a potential role in POAG and is mentioned in our review (lines 299-203, 246-251 & especially lines 291-308 and also in our concluding remarks lines 353-367). Mir-29b was presented in table 1 & figure 1

Is there any data which suggests that this miRNA could be initially abnormal in the POAG?

During the last years researchers have been looking for miRNAs as bio markers for different glaucoma diseases by comparing the levels of different miRNAs in patient's plasma and AH with promising results (Hindle, Allyson G., et al IOVS 60.1 (2019): 134-146, Tanaka, Yuji, et al. Sci reports 4.1 (2014): 1-7.and other.‏) Although the term "abnormal miRNA" was used regarding different diseases, we could not find any publication discussing the presence and involvement of abnormal miRNAs in POAG. We believe that the presence of abnormal miRNA is a reasonable possibility and that abnormal miRNA discoveries may shed light on some of the unclear pathological processes taking place in POAG.

Reviewer 3 Report

Tabak and coauthors submitted a paper describing the association of microRNAs and oxidative stress in the pathogenesis oh POAG.  The authors did a comprehensive review of the role of microRNAs in POAG, especially in the processes of inflammation and oxidative stress. I have no further comment on the paper.  However, there are several grammatical and spelling errors throughout the article. In addition, the authors used many long sentences to describe a complicated process.  Therefore, dividing a long sentence into two short sentences is suggested. The readers will be more easier to understand the complicated physiological and pathological processes.

Author Response

N/A

Round 2

Reviewer 2 Report

I would like to thank the authors for the modifications and improvement of their text.

It is more clear as presented here and it can be accepted as it is for publication.